# Rapid Detection of HLA-B*57:01-Expressing Cells Using a Label-Free Interdigitated Electrode Biosensor Platform for Prevention of Abacavir Hypersensitivity in HIV Treatment

**DOI:** 10.3390/s19163543

**Published:** 2019-08-14

**Authors:** Jianxiong Chan, Gita V. Soraya, Lauren Craig, Shah M. Uddin, Marian Todaro, Duc H. Huynh, Chathurika D. Abeyrathne, Lyudmila Kostenko, James McCluskey, Efstratios Skafidas, Patrick Kwan

**Affiliations:** 1Department of Medicine, Royal Melbourne Hospital, The University of Melbourne, Victoria 3050, Australia; 2Department of Biochemistry, Faculty of Medicine, Hasanuddin University, Makassar 90245, Indonesia; 3Department of Neurology, Royal Melbourne Hospital, Victoria 3050, Australia; 4Department of Electrical and Electronic Engineering, Melbourne School of Engineering, The University of Melbourne, Victoria 3010, Australia; 5Department of Microbiology and Immunology, The Peter Doherty Institute for Infection and Immunity, The University of Melbourne, Victoria 3010, Australia

**Keywords:** human leukocyte antigen, interdigitated electrodes, impedance sensor, protein sensor, point-of-care diagnostics, pharmacogenetics

## Abstract

Pre-treatment screening of individuals for human leukocyte antigens (HLA) HLA-B*57:01 is recommended for the prevention of life-threatening hypersensitivity reactions to abacavir, a drug widely prescribed for HIV treatment. However, the implementation of screening in clinical practice is hindered by the slow turnaround time and high cost of conventional HLA genotyping methods. We have developed a biosensor platform using interdigitated electrode (IDE) functionalized with a monoclonal antibody to detect cells expressing HLA-B*57:01. This platform was evaluated using cell lines and peripheral blood mononuclear cells expressing different HLA-B alleles. The functionalized IDE sensor was able to specifically capture HLA-B*57:01 cells, resulting in a significant change in the impedance magnitude in 20 min. This IDE platform has the potential to be further developed to enable point-of-care HLA-B*57:01 screening.

## 1. Introduction

Adverse drug reactions (ADRs) account for more than 700,000 injuries or deaths per year in the United State alone [1]. Recent pharmacogenetics research has identified associations between the human leukocyte antigens (HLA) encoded by the major histocompatibility complex (MHC) gene family and ADRs. These immune-mediated ADRs manifest as cutaneous reactions which range from minor maculopapular exanthema to severe reactions such as Stevens Johnson syndrome (SJS), toxic epidermal necrolysis (TEN), and drug-induced hypersensitivity syndrome (DIHS). These severe reactions carry mortality rates of up to 30% [2]. The MHC gene family is a highly polymorphic region in the human genome and is known for its role in immune response and various disease processes. A high level of variations is found within the Class I HLA region, which encompasses the HLA-A, HLA-B, and HLA-C genes. There are currently 12,544 allele sequences within Class I recorded in the International Immunogenetics (IMGT)/HLA database, of which 4828 belong to the HLA-B group [3].

A prime example of such pharmacogenetic association is the strong relationship between HLA-B*57:01 and a hypersensitivity reaction to abacavir [4,5,6,7]. This association appears to be strictly linked to HLA-B*57:01 and not to other closely related HLA allotypes [8,9]. Abacavir is widely prescribed for HIV treatment, but approximately 5% of patients develop hypersensitivity reactions [5]. The most common symptoms are fever, rash, fatigue and nausea or vomiting, while respiratory and cardiac symptoms have also been reported [5]. HLA testing prior to abacavir administration is recommended by the U.S. Food and Drug Administration (FDA) and is supported by expert opinions [10,11,12,13].

HLA-B*57:01 is traditionally detected by DNA-based methods such as polymerase chain reaction (PCR) and sequence-based typing. These laboratory-based methods are time consuming and costly, which impair the implementation of HLA testing [14,15,16]. This delay is also impractical when a treatment decision is preferred to be made instantaneously for optimal care or in resource-poor environments.

Our group has previously developed a monoclonal antibody (mAb), known as 3E12, capable of recognizing members of the HLA-B17 group, including HLA-B*57:01 [17]. The sensitivity and specificity of this antibody was demonstrated in three independent laboratories using flow cytometry [17]. In those studies, it was shown that patients who tested negative by mAb screening comprised 90–95% of all individuals in most ethnic populations and required no further HLA typing [17]. This mAb screening may provide a low-cost alternative to high-resolution typing of all patients and rapid ascertainment of low-risk patients who can begin immediate therapy with abacavir.

The present study aimed to develop this mAb from a laboratory-based application towards a biosensor platform. We combined the selectivity of this mAb with sensitivity of biosensor, setting the basis for future point-of-care application. Biosensors are promising options due to their high sensitivity, low cost, and amenability to miniaturisation. Through a point-of-care application, samples would not need to be sent to distant laboratories, greatly reducing the time and cost for HLA-B*57:01 screening.

## 2. Materials and Methods

### 2.1. Overall Design

The central design of the platform comprises of an interdigitated electrode (IDE) sensor functionalized with an antibody targeting against specific cell surface protein. The IDE sensors were fabricated via UV lithography at low cost and have previously demonstrated sensitivity for label-free detection of various biomolecules [18,19,20,21,22].

For HLA-B*57:01 cell detection, the mAb 3E12 was immobilized on the IDE sensor surface. Cell overexpressing HLA-B*57:01 protein on the surface bound to 3E12 on the sensor. The binding of cells to the antibody resulted in changes in signal impedance of the IDE sensor. The changes in signal impedance were detected by a lock-in amplifier (Figure 1).

A cross-sectional view of the measurement setup illustrates detection of captured HLA-B*57:01 cells in a sample medium with the sensing area. Probe electrodes were placed to deliver excitation current to and to measure electrical signals from the sensors.

There were two experimental stages in this study. Stage I involved proof-of-concept experiments followed by optimisation of the experimental parameters using cultured cell lines with and without HLA-B*57:01 expression. The cell capture conditions were optimized on a glass surface followed by signal characterisation of the capture on an IDE sensor. The parameters developed in stage I were evaluated in stage II using human peripheral blood mononuclear cells (PBMC) of known HLA genotypes. The study was approved by The University of Melbourne ethics committee (ethics ID 1443204.4).

### 2.2. Samples

#### 2.2.1. Cell Lines

Human lymphoblastoid HLA Class I-reduced cell lines (C1R) overexpressing HLA-B*57:01 or HLA-B*15:02, were cultured in Roswell Park Memorial Institute (RPMI) media (Thermo Fisher Scientific, MA, USA) supplemented with 2 mM Minimum Essential *Medium* (MEM) non-essential amino acid solution (Thermo Fisher), 100 mM (4-(2-hydroxyethyl)-1-piperazineethanesulfonic acid (HEPES) (ICN Biochemicals, Aurora, OH, USA), 2 mM l-glutamine (Merck, Darmstadt, Germany), 0.6 mg/L benzylpenicillin (CSL, Melbourne, Australia), 1 mg/L streptomycin (CSL), 50 µM 2-mercaptoethanol (Merck) and 10% FBS (CSL, Melbourne) with 1% G418 selection antibiotic (Thermo Fisher) and were kept in humid conditions at 37 °C, 5% CO_2_.

#### 2.2.2. Isolated PBMC Samples

PBMCs were prepared from blood samples of healthy donors and isolated via the standard Ficoll-Paque and density-gradient centrifugation method. Isolated cells were stored in liquid nitrogen until use. Samples were rapidly thawed with RPMI media, pelleted and supernatant removed. Cells were resuspended in RPMI (10% FCS) and kept in humid conditions at 37 °C, 5% CO_2_.

### 2.3. Antibody

The mAb, 3E12, was functionalized on glass and an IDE sensor. The antibody was previously produced by our group [17] against the HLA-B17 serotype, which comprises of B*57:01 and B*58:01.

### 2.4. Functionalization of Glass Surface

To attach the 3E12 antibody, the glass slides were first cleaned with ethanol and acetone at a 50:50 ratio, sonicated for 5 min and air dried. Slides were incubated for 1 h in 2% (3-aminopropyl) triethoxysilane (APTES) (Merck) in 95% ethanol, washed in 1% Phosphate Buffered Solution (PBS) and incubated in 2.5% glutaraldehyde (Merck) in distilled water for 2 h. Following incubation, the slides were washed in distilled water and the antibodies were immobilized at 100 µg/mL on the surface overnight at 4 °C. After incubation, the slides were washed with PBS and incubated for 1 h in 1% ethanolamine. Finally, the slides were washed and kept in PBS.

### 2.5. Fabrication and Functionalization of IDE Sensor

#### 2.5.1. Sensor Fabrication

The IDE sensors were fabricated at the Melbourne Centre for Nanofabrication (Clayton, Victoria, Australia) using UV-lithography on BOROFLOAT glass wafers [18,19,20,21]. Each IDE sensor consisted of paired electrode arrays with a finger length of 980 µm, a finger width of 8 µm and a gap width of 8 µm. An additional level of SiO_2_ (25 nm thickness) was deposited through e-beam evaporation.

#### 2.5.2. Sensor Functionalization

The fabricated sensors were washed with acetone, isopropyl alcohol and H_2_O, and dried under nitrogen gas. Sensors were then plasma treated (PE-25 Plasma Etch, NV, USA) with argon (75%) and oxygen (25%) for 5 min at 50 W power, and 30 cc/min flow rate. The sensors were then functionalized in accordance to our established protocols [18,19,20,21]. Briefly, sensors were incubated in filtered (Corning^®^ 0.2 μm) 2% APTES in ethanol solution for 1 h, followed by 3 × 5 min washes in 100% ethanol with gentle shaking. Sensors were then incubated in filtered 2.5% glutaraldehyde solution for 2 h, followed by 3 × 5 min wash in PBS with gentle shaking and dried under nitrogen gas. Antibodies were then incubated on sensing area at 100 µg/mL at 4 °C overnight. After incubation, the slides were washed with PBS and incubated for 1 h in 1% ethanolamine before washed and stored in PBS.

### 2.6. Cell Capture Assay

The cell lines or PBMCs, depending on experimental stages, were incubated at 160 cells/µL in a glass or sensor surface in their respective culture media at 37 °C. After the applicable incubation time, the slides or sensors were washed three times in PBS and counted under a microscope in PBS.

### 2.7. Cell Counting

Images of slides or sensors with captured cells were taken at 50× magnification by inverted microscope using Axio Image software. Cell counts were performed on an area of 0.5 mm^2^ using ImageJ software, utilizing threshold adjustment and article analysis tools.

### 2.8. Electrical Measurement

Impedance of the sensors was measured using established circuit setup [18,19,20,21], in which the sensor, represented as a resistor (R), was in a series with a capacitor (*C*) connected in series with a 1 kΩ reference resistor (*R_ref_*). The input sinusoidal AC excitatory signal (*V_in_*) was provided by a function generator at 20 mV peak-to-peak voltage (*V_pp_*) using different frequencies (10 kHz, 20 kHz, 40 kHz and 60 kHz) as required. Sensors were first measured in wet-state (in PBS) for 2 time points, firstly for a baseline measurement (before sample incubation) and then after sample incubation and washing. The amplitude of the output voltage (*V_out_*) and the phase across the *R_ref_*_,_ were recorded by a lock-in amplifier. This was used for the acquisition of frequency (*ω*) dependent impedimetric parameters such as impedance magnitude (|Z|), capacitance (*C*), and resistance (*R*) based on the following equation [20,21]:*V_out_*/*V_in_* = *R_ref_*/(|Z| + *R_ref_*)(1)
|Z| = R − j/(*wC*)(2)

The measurement protocol was first characterized in stage I and then applied in stage II.

### 2.9. Statistical Analysis

The number of surface-bound cells are expressed as mean ± standard error of mean (mean ± SEM). MATLAB^®^ was used to calculate the sensor impedance magnitude. The baseline impedance values were used to determine outliers in accordance with our previously developed method [21]. Results were described as a percentage of change in impedance magnitude (%∆|Z|) before and after incubation. This was used throughout the results section, unless otherwise stated. Statistical analysis was performed by GraphPad Prism 7.0 software. Non-parametric Kruskal–Wallis one-way ANOVA followed with Dunn’s multiple comparison tests were performed on data with sample sizes less than 10. For grouped samples, non-parametric two-way ANOVA with Bonferroni correction for multiple comparisons was used. *p*-values ≤ 0.05 were considered statistically significant.

## 3. Results

### 3.1. Stage I: Specific Cell Capture on Glass and IDE Sensors Using Cell Lines

#### 3.1.1. Feasibility of Cell Capture on Glass

HLA-B*57:01 expressing cell line was incubated on glass surfaces functionalized with a 3E12 antibody or surfaces without an antibody, at 160 cells/µL for 10 min at 37 °C. As observed in Figure 2A, a higher number of cells were captured on the surface with a 3E12 antibody compared to the surface without the antibody where very few cells were captured. To determine if this capture was selective for HLA-B*57:01, glass surfaces with or without a functionalized 3E12 antibody were incubated with either HLA-B*57:01 or HLA-B*15:02 cell lines. As shown in Figure 2B, the surface functionalized with 3E12, when incubated with HLA-B*57:01 cell line, had the highest number of cells captured. This capture was significantly higher compared to other conditions with the HLA-B*15:02 cell line, or to the control surface without antibody.

#### 3.1.2. Cell Capture over Time

The minimum time required for cell capture was evaluated. HLA-B*15:02 cell line and glass without functionalized antibodies were used as controls. Glass surfaces were incubated in HLA-B*57:01 or HLA-B*15:02 cell lines for 5, 10 and 20 min at a concentration of 160 cells/µL (1 mL each, 160,000 cells) at 37 °C. As shown in Figure 3A, on an area of 0.5 mm^2^, the number of HLA-B*57:01 positive cells captured on the functionalized surface started to rise after incubation for 10 min and continued to rise after incubation for 20 min. At 20 min of incubation, 314 ± 24 (mean ± SEM) HLA-B*57:01 cells were captured compared to 55 ± 9 HLA-B*15:02 cells (*p* = 0.0004). At these time points the number of cells captured was significantly higher compared to the HLA-B*15:02 cell line and surface without functionalized antibodies, for which the number of cells captured remained low throughout the incubation time period (Figure 3B).

#### 3.1.3. IDE Sensor Detection of HLA-B*57:01 Cell Line

Next, we determined whether the cell capture can be detected on the IDE sensors. 3E12 antibody was functionalized on IDE sensors and incubated with either HLA-B*57:01 expressing cells or HLA-B*15:02 expressing cells at 37 °C for 20 min, the time point with maximum capture difference (Figure 3B). Sensors incubated with HLA-B*57:01 cell lines showed a higher number of cells captured on the sensors surface compared to sensors with HLA-B*15:02 expressing cells (Figure 4A).

This capture was then subject to an electrical reading (Figure 4B) to detect difference in percentage changes in impedance magnitude (%∆|Z|) of the sensors with captured cells from HLA-B*57:01 positive and negative cell lines. Then four frequencies, 10 kHz, 20 kHz, 40 kHz and 60 kHz were applied for the measurement. It was observed that 10 kHz was the optimal frequency to differentiate the sensors with captured cells. At 10 kHz, the sensors incubated with HLA-B*57:01 positive cells showed an increase in impedance magnitude. By contrast, sensors incubated with negative controls (HLA-B*15:02 and media control) generally showed decrease in impedance magnitude.

### 3.2. Stage II: Detection of Peripheral Blood Mononuclear Cells

In this part of the study, the protocol developed in stage I was evaluated using PBMCs isolated from whole blood from donors of known HLA genotypes. Similar to the cell line experiments, glass surfaces were used followed by IDE sensors.

#### 3.2.1. Specific PBMC Capture on Glass

Glass surfaces functionalized with 3E12 antibody were incubated with 1 mL of either HLA-B*57:01 positive or negative PBMCs (160 cells/µL) for 20 min at 37 °C. Surfaces functionalized with a 3E12 antibody had significantly greater number of captured cells when incubated with HLA-B*57 positive PBMCs (HLA-B*57 positive and HLA-B*57 negative capture: Mean ± SEM 777 ± 78 and 117 ± 13 respectively, Figure 5). The number of cells captured with HLA-B*57-negative PBMCs are similar to the background and negative controls (Figure 5C). This suggested that surfaces functionalized with 3E12 antibody selectively captured PBMC expressing HLA-B*57 positive and not negative cells.

#### 3.2.2. Sensor Detection of HLA-B*57:01 Positive PBMCs

Next, we determined whether the capture of PBMCs can be detected on the IDE sensors. IDE sensors functionalized with 3E12 antibody were incubated with 10 different PBMC samples (five HLA-B*57:01 positive, five negative, 1 mL each at 160 cells/µL) for 20 min at 37 °C. Sensors incubated with HLA-B*57:01 positive PBMCs captured more cells on their surfaces compared with sensors incubated with HLA-B*57 negative PBMCs and media controls (Figure 6A). This difference in capture was reflected in the impedance (%∆|Z|) changes of sensors (Figure 6B). To compare the %∆|Z| values, sensor measurements were combined into groups of HLA-B*57:01 positive, negative, and media control incubated (Figure 6C). Sensors incubated with positive PBMCs showed an average impedance magnitude change of −4% ± 1.96%, which is highly distinct from the average signal of sensors incubated with negative PBMC at 0.74% ± 2.4% (*p* < 0.0001), and blank media at −0.58% ± 1.21% (*p* = 0.02). This suggested that the IDE sensors were able to differentiate the capture of PBMCs of different HLA types.

## 4. Discussion

This study presented a novel way of HLA-B*57:01 testing through antibody capture of HLA-B*57:01 expressing cells and electrical detection by IDE sensors. Results were obtained in 20 min of cell incubation at 37 °C.

The first stage of the study involved proof-of-concept and optimisation experiments. The feasibility and the characteristics of cells captured were first established. This was achieved through a series of proof-of-concept experiments conducted on silicon-dioxide (glass) surfaces using HLA-B*57:01 expressing and non-expressing cell lines. The glass slides that were used as the surface have similarities with the IDE sensor and its conditions can be readily manipulated when required. As seen in Figure 2, glass surfaces functionalized with a 3E12 antibody was able to capture whole cells expressing HLA-B*57:01 and did not capture HLA-B*57:01 negative cells lines after incubating for 10 min to 20 min.

The cell capture condition was then applied to functionalized IDE sensors where captured cells underwent electrical detection to characterize its impedance magnitude, a method of detection that is suitable for adaptation into a point-of-care device. It was observed that the capture of HLA-B*57:01 cells lines resulted in an increase of the sensors’ impedance magnitude. In contrast, incubation with HLA-B*57:01 negative cells line and media resulted in a decrease of the sensors’ impedance magnitude. This suggested that cell capturing by the IDE sensor could be detected by determining the change in impedance. The experimental conditions established in this part of the study formed the basis for stage II where they were extended to evaluate PBMCs isolated from human blood samples.

In stage II, glass surface functionalized with 3E12 showed similar capture characteristics of PBMCs to cell lines in stage I. A higher number of HLA-B*57 positive PBMCs was captured compared to HLA-B*57 and B*57 negative PBMCs. The PBMC samples were then tested on IDE sensors functionalized with a 3E12 antibody. It was found that these sensors were able to capture more HLA-B*57:01 positive PBMCs compared to HLA-B*57 negative samples, resulting in differences in impedance change. It was noted that the characteristic of change in impedance using PBMCs was different from that observed using cell lines. This could be due to the difference in the cell types of the samples. PBMC contains many different types of white cells, whereas the cell line used is derived from one cell type within PBMC (B-cell), which makes up of less than 10% of all the cell types [23,24]. Sample dependent differences in sensitivity in impedance sensors have been observed previously [25]. Furthermore, cell lines are generally actively dividing while PBMCs are not. This might result in a difference in ionic composition of the media between the cell lines and PBMCs, resulting in different characteristics of impedance change.

Since impedance-based detection detects the changes of conductivity resulting from the differences between electrical property of the target and its medium [26], it is important to perform characterisation for the specific type of sample used. Despite the difference in the pattern of change, the frequency of 10 kHz allowed a highly specific detection of capture cells across both sample types, without the use of additional signal labelling or amplification strategies used in other impedance-based HLA-B detection previously reported [20].

Our approach has its limitations. In this study, pre-treated blood samples (PBMCs) were used to characterize cell capture on the sensor. As complex pre-treatment of blood is not be ideal for point-of-care setting, future work to characterize the sensor using untreated whole blood samples of different HLA groups are needed. To bring this work closer towards point-of-care settings, further work on the essential sensor’s performance will be needed. These will include establishing the limit of detection to establish an optimal detectable cell capture range, sensor stability, sensitivity and reproducibility under different conditions and environmental factors.

Though the mAb 3E12 is specific to the B17 group it does not discriminate between the subtypes within this group, which include HLA-B*57:01 or HLA-B*58:01 [11]. The carrier rate of HLA-B*57:01 varies across ethnic populations, from less than 1% in African population to 20% in parts of India [11]. In comparison, HLA-B*58:01 is mostly found in Asian subpopulations, particularly in individuals of Korean, Han Chinese, or Thai descent with carrier rate that varies across populations, from less than 6% in the Korean population to 17% in parts of China [27,28,29,30]. Therefore, although our HLA-B*57 sensing platform does not uniquely detect HLA-B*57:01, it can be used to exclude most individuals from costly HLA testing and allow them to receive abacavir with confidence. In other words, only patients who test positive would need to undergo gold standard HLA-B*57:01 genotyping and withhold prescription of abacavir until this result is known. This practice would be able to reduce the overall cost and time constraints of screening. Furthermore, HLA-B*58:01 is associated with hypersensitivity to allopurinol, which is commonly used to treat hyperuricaemia and recurrent gout [31,32]. Therefore, future studies could explore the potential application of the IDE sensors for pre-treatment screening prior to allopurinol use.

## 5. Conclusions

We have developed an IDE sensor platform for label-free HLA-B*57:01 screening for the prevention of abacavir hypersensitivity. This was done using the 3E12 monoclonal antibody -functionalized IDE sensors. Specific HLA-B*57:01 PBMC capture was attained in just 20 min, with a significant change in the impedance magnitude detected by the sensors. With further development, this platform can be adapted to a point-of-care system which can facilitate HLA testing for ADR from abacavir use in HIV treatment.

## Figures and Tables

**Figure 1 sensors-19-03543-f001:**
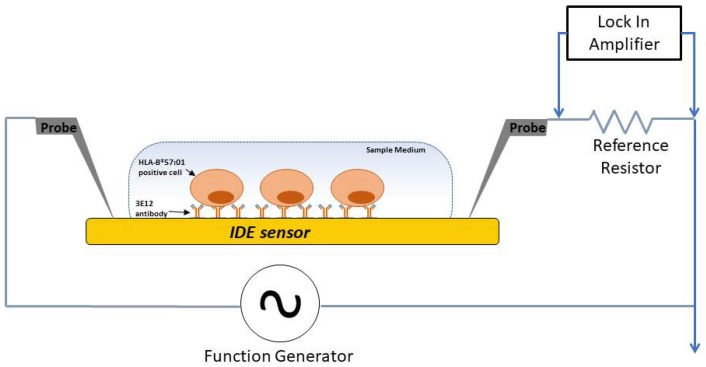
Circuit model and measurement setup used for detection of HLA-B*57:01 cell capture.

**Figure 2 sensors-19-03543-f002:**
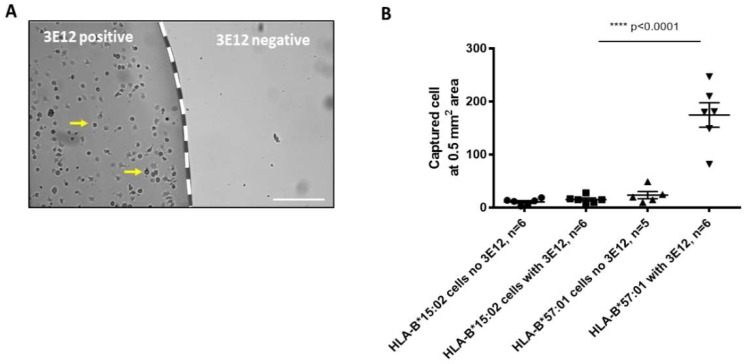
A 3E12 antibody capture HLA-B*57:01 expressing cell line on glass surface. (**A**) Glass surface showing HLA-B*57:01 positive cells captured on glass surface (yellow arrow) functionalized with a 3E12 antibody (left) and not on surface without a 3E12 antibody (right). (**B**) Significantly higher number of cells were captured when HLA-B*57:01 positive cells were incubated with glass surfaces functionalized with a 3E12 antibody compared to other HLA groups (HLA-B*15:02) and controls within an area of 0.5 mm^2^ (mean ± SEM. Bar = 100 µm).

**Figure 3 sensors-19-03543-f003:**
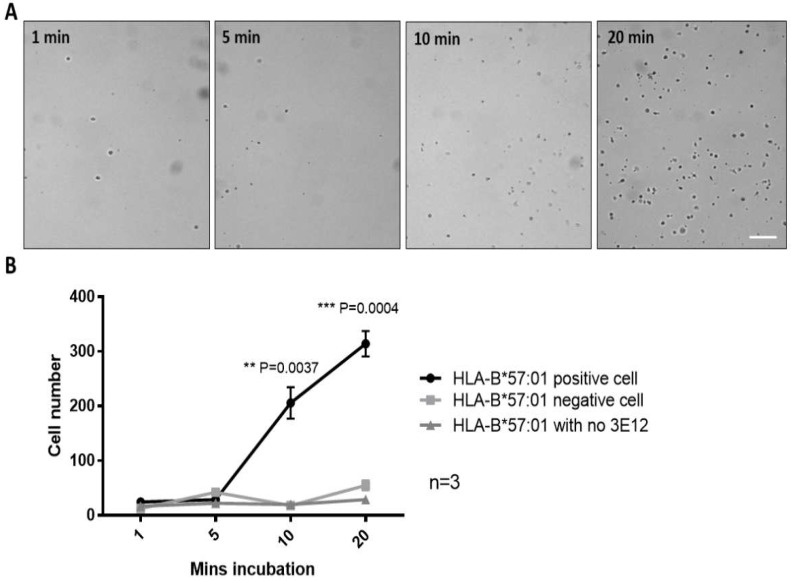
HLA-B*57:01 expressing cell line capture over time. (**A**) HLA-B*57:01 positive cells captured on a glass surface functionalized with a 3E12 antibody at 1 min, 5 min, 10 min and 20 min of incubation. (**B**) A Significant increase in the number of HLA-B*57:01 positive cells were captured from 10 min of incubation compared to HLA-B*57:01 negative cells and no antibody control at 0.5 mm^2^ area (mean ± SEM. Bar = 100 µm).

**Figure 4 sensors-19-03543-f004:**
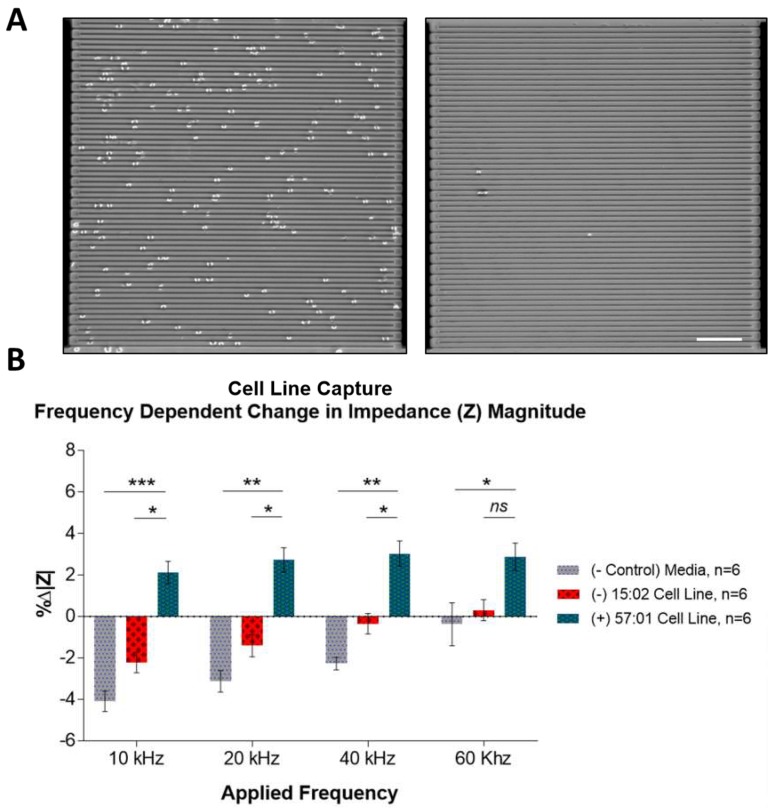
Interdigitated electrode (IDE) sensors detecting sensors with captured HLA-B*57:01 expressing cell line. (**A**) Micrograph showing HLA-B*57:01 positive cells captured on dielectric sensors functionalized with 3E12 antibody (left). Less cells were captured when HLA-B*15:02 cells were incubated on the sensor (right). (**B**) At applied frequency of 10 kHz and 20 kHz, sensors showed significant impedance difference with sensors incubated HLA-B*57:01 cells compared to HLA-B*15:02 cells and blank. (mean ± SEM. Bar = 100 µm).

**Figure 5 sensors-19-03543-f005:**
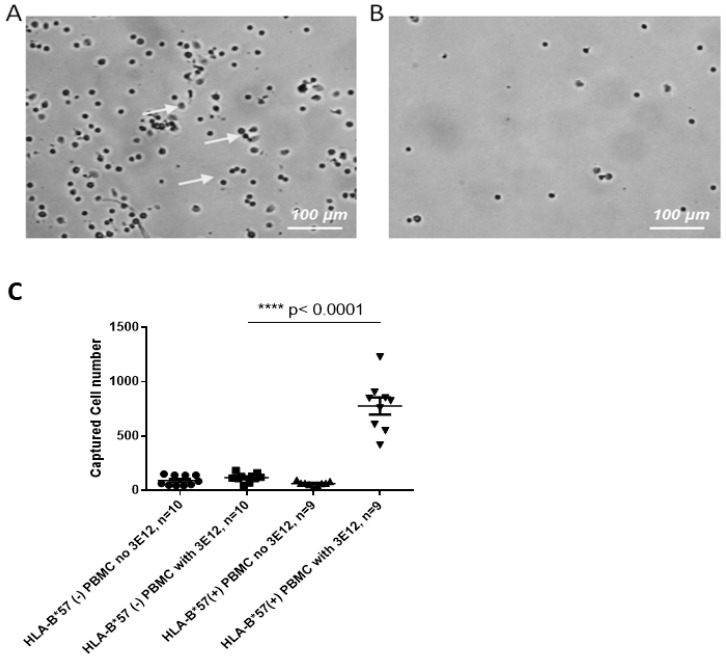
A 3E12 antibody capture HLA-B*57 expressing peripheral blood mononuclear cells (PBMCs) on glass surface. Glass surface showing more HLA-B*57 PBMCs captured on glass surface (white arrow) functionalized with a 3E12 antibody (**A**) compared to the surface without 3E12 antibody (**B**). (**C**) Significantly higher number of cells were captured when HLA-B*57 PBMCs were incubated with glass surfaces functionalized with a 3E12 antibody compared to other HLA group and controls at 0.5 mm^2^ area (mean ± SEM. Bar = 100 µm).

**Figure 6 sensors-19-03543-f006:**
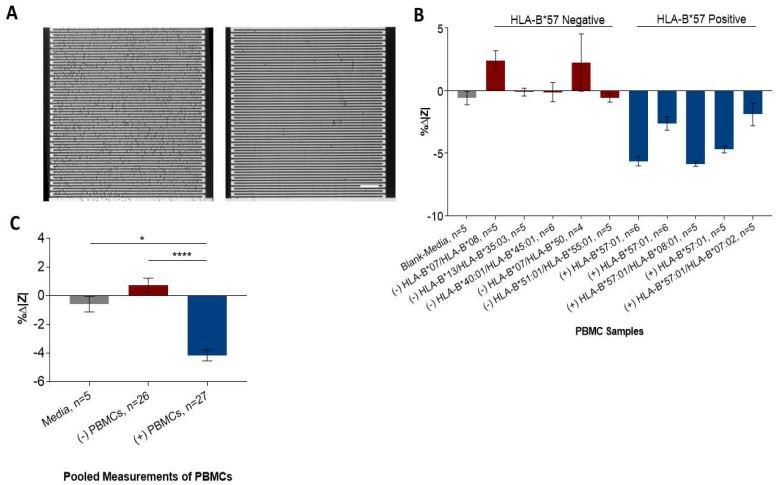
IDE sensors detecting sensors with captured HLA-B*57 expressing PBMCs. (**A**) Micrograph showing HLA-B*57:01 PBMC captured on dielectric sensors functionalized with a 3E12 antibody (left). Less cells were captured when non-HLA-B*57:01 were incubated on the sensor (right). (**B**) At applied frequency of 10 kHz, sensors showed significant impedance difference with sensors incubated HLA-B*57:01 PBMCs cells compared to non-HLA-B*57:01 PBMCs. (**C**) Sensor measurements were combined into groups of HLA-B*57:01 positive, HLA-B*57:01 negative, and media control incubated (mean ± SEM. Bar = 100 µm).

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
