# Peer review of "Rapid Detection of HLA-B*57:01-Expressing Cells Using a Label-Free Interdigitated Electrode Biosensor Platform for Prevention of Abacavir Hypersensitivity in HIV Treatment"

_sensors, 2019, doi:10.3390/s19163543_

Round 1

Reviewer 1 Report

This manuscript presents a label-free electrode biosensor platform for detection of HLA-B*57:01. In the presence of positive cells, they can be captured by the sensor, leading to a change in the impedance magnitude. In summary, this manuscript would be publishable if the following questions can be addressed properly.

1) It was published by the same authors that an electrode biosensor for sensing HLA-B*15:02 genotyping for prevention of drug hypersensitivity (Biosensors and Bioelectronics 111 (2018) 174–183). Therefore, it is recommended to address the novelty of the current work compared with the published work.

2) How is the performance of the developed biosensor in real blood samples? It is not a POC platform if complex sample pre-treatment is required.

3) It is difficult to distinguish the columns in Figure 4B when the manuscript is printed in black and white, it is recommended to add patterns into the columns.

Author Response

Point 1: It was published by the same authors that an electrode biosensor for sensing HLA-B*15:02 genotyping for prevention of drug hypersensitivity (Biosensors and Bioelectronics 111 (2018) 174–183). Therefore, it is recommended to address the novelty of the current work compared with the published work.

Response 1: The novelty of current work is that it involves direct specific cell detection on sensor surface to test for HLA-B*57:01. This is different to our previous work where amplified HLA-B*15:02 gene was detected instead. The previous work will require additional step of DNA amplification before detection on sensor. The following sentence has been modified to address this point:

L331-334: Despite the difference in the pattern of change, the frequency of 10 kHz allowed highly specific detection of capture cells across both sample types, without use of additional signal labelling or amplification strategies used in other impedance-based HLA-B detection previously reported [20]

Point 2: How is the performance of the developed biosensor in real blood samples? It is not a POC platform if complex sample pre-treatment is required.

Response 2:  We agree that it would be important to use whole blood samples for the eventual point-of-care application. However, this study focused on testing the biosensor on cell lines and peripheral blood mononuclear cells (PBMC) because the cell component is essential for the initial characterizing of the cell capture ability of the sensor.  This is a limitation of this study and whole blood samples should be investigated in future work. The following paragraph is added to emphasize this point:

L337-340: Our approach has limitations. In this study, pre-treated blood samples (PBMCs) were used to characterize cell capture on the sensor. As complex pre-treatment of blood is not be ideal for point-of-care setting, future work to characterise the sensor using untreated whole blood samples of different HLA groups is needed.

Point 3: It is difficult to distinguish the columns in Figure 4B when the manuscript is printed in black and white, it is recommended to add patterns into the columns.

Response 3: Patterns have been added to distinguish the columns in Figure 4B

Reviewer 2 Report

In this paper, the author described a biosensor platform using IDE functionalized with the 3E12 mAb to detect cells expressing HLA‐B*57:01. They reported that the functionalized IDE sensor was able to specifically capture HLA‐B*57:01 cells, resulting in a significant change in the impedance magnitude in 20 minutes. I agree that it is important to develop the point-of-care HLA‐B*57:01 screening. But, it is not suitable to be published on this journal in its present form. There are some issues that I feel need addressing.

1) The authors should be presented the essential sensing performance of their IDE sensor including sensitivity, reproducibility and stability, etc.. How long can the capture ability of IDE sensor functionalized with 3E12 mAb be maintained?

2) Part 3.1.2. – Please add the cell number or sample volume of HLA‐B*15:02 cell line, as well as incubation temperature. What is the capture efficiency of glass surface functionalized with 3E12 mAb at 20 min incubation?   

3) What is the total sample volume of PBMCs for IDE sensor and/or glass? The authors described that glass surfaces functionalized with 3E12 Ab were incubated with HLA‐B*57:01 positive or negative PBMCs (160 cells/μl). What is the capture efficiency of glass surface functionalized with 3E12 mAb?

4) What is the low detection limit of IDE sensor? Can the IDE sensor identify the capture of lower number of positive PBMCs, through the change in the impedance magnitude?   

5) What do "the yellow arrow in Fig. 2(A)" and "white arrow in Fig. 5(A)" indicate?

Author Response

Point 1: The authors should be presented the essential sensing performance of their IDE sensor including sensitivity, reproducibility and stability, etc.. How long can the capture ability of IDE sensor functionalized with 3E12 mAb be maintained?

Response 1: The current work is a proof-of-concept study where the focus was to establish the cell capture ability rather than the stability and performance of the sensor. The following paragraph is added to the limitation and future work section to emphasize on this:

L340-343: To bring this work closer towards point-of-care settings, further work on the essential sensor’s performance will be needed. These will include establishing the limit of detection to establish an optimal detectable cell capture range, sensor stability, sensitivity and reproducibility under different conditions and environmental factors.

Point 2.1: Part 3.1.2. – Please add the cell number or sample volume of HLA‐B*15:02 cell line, as well as incubation temperature.

Response 2.1: The following line has been added:

L210-212: Glass surfaces were incubated in HLA-B*57:01 or HLA-B*15:02 cell lines for 5, 10 and 20 minutes at concentration of 160 cells/µl (1 ml each, 160,000 cells) at 37oC.

Point 2.2: What is the capture efficiency of glass surface functionalized with 3E12 mAb at 20 min incubation?   

Response 2.1: At 20 mins incubation, within an area of 0.5 mm2, there were 314±24 HLA-B*57:01 cells captured (mean±SEM) as compared to 55±9 HLA-B*15:02 cells captured (mean±SEM). The following sentence has been added to clarify this point:

L212-215: As shown in Figure 3A, on an area of 0.5 mm2, the number of HLA-B*57:01 positive cells captured on the functionalised surface started to rise after incubation for 10 minutes and continued to rise after incubation for 20 minutes. At 20 minutes of incubation, 314±24 (mean±SEM) HLA-B*57:01 cells were captured compared to 55±9 HLA-B*15:02 cells (p=0.0004).

Point 3.1: What is the total sample volume of PBMCs for IDE sensor and/or glass?

Response 3.1: The volume of PBMCs used for both sensors and glass were 1 ml (160 cells/μl) per incubation. The following lines has been modified to clarify this point:

L255-256: Glass surfaces functionalised with 3E12 antibody were incubated with 1 ml of either HLA-B*57:01 positive or negative PBMCs (160 cells/µl) for 20 minutes at 37oC.

L270-272: IDE sensors functionalised with 3E12 antibody were incubated with 10 different PBMC samples (5 HLA-B*57:01 positive, 5 negative, 1 ml each at 160 cells/µl) for 20 minutes at 37oC.

Point 3.2: The authors described that glass surfaces functionalized with 3E12 Ab were incubated with HLA‐B*57:01 positive or negative PBMCs (160 cells/μl). What is the capture efficiency of glass surface functionalized with 3E12 mAb?

Response 3.2: The capture efficiency within an area of 0.5 mm2 after 20 min incubation was 777±78 (mean±SEM) cells for HLA-B*57:01 positive PBMCs compared to 117±13 cells for HLA-B*57:01 negative PBMCs. The following sentence has been added to clarify this point:

L256-259: Surfaces functionalised with 3E12 antibody had significantly greater number of captured cells when incubated with HLA-B*57 positive PBMCs (HLA-B*57 positive and HLA-B*57 negative capture: mean±SEM 777±78 and 117±13 respectively, Figure 5).

Point 4: What is the low detection limit of IDE sensor? Can the IDE sensor identify the capture of lower number of positive PBMCs, through the change in the impedance magnitude?   

Response 4: The low detection limit of IDE the sensor was not determined as the focus of this study was to establish the cell capture ability and selectivity. We acknowledge this limitation and agree that it would important to determine the lower detection limit in future work. The following paragraph is added in the limitation and future direction section:

L340-343: To bring this work closer towards point-of-care settings, further work on the essential sensor’s performance will be needed. These will include establishing the limit of detection to establish an optimal detectable cell capture range, sensor stability, sensitivity and reproducibility under different conditions and environmental factors.

Point 5: What do "the yellow arrow in Fig. 2(A)" and "white arrow in Fig. 5(A)" indicate?

Response 5: The yellow arrow in Figure 2A indicated captured cells on the glass slide. The white arrow in Figure 5A indicated captured PBMCs on the glass slide. The following Figure legends (Figure 2 and 5) were modified to clarify this point:

L201-203: (A) Glass surface showing HLA-B*57:01 positive cells captured on glass surface (yellow arrow) functionalised with 3E12 antibody (left) and not on surface without 3E12 antibody (right).

L264-266 Glass surface showing more HLA-B*57 PBMCs captured on glass surface (white arrow) functionalised with 3E12 antibody (A) compared to the surface without 3E12 antibody (B).

Round 2

Reviewer 1 Report

The authors addressed the issues properly. This manuscript can be accepted.

Reviewer 2 Report

The manuscript was modified as suggested.